# An Improved Multi-Object Tracking Algorithm Designed for Complex Environments

**DOI:** 10.3390/s25175325

**Published:** 2025-08-27

**Authors:** Wuyuhan Liu, Jian Yao, Feng Jiang, Meng Wang

**Affiliations:** 1School of Electronic Information and Physics, Central South University of Forestry and Technology, Changsha 410004, China; sixfive6655@163.com (W.L.); t20040535@csuft.edu.cn (F.J.); 2School of Computer and Information Engineering, Central South University of Forestry and Technology, Changsha 410004, China; t20110565@csuft.edu.cn

**Keywords:** MOT, JDE, reparameterization, attention mechanism, RGTrack

## Abstract

Multi-object tracking (MOT) algorithms are a key research direction in the field of computer vision. Among them, the joint detection and embedding (JDE) method, with its excellent speed and accuracy performance, has become the current mainstream solution. However, in complex scenes with dense targets or occlusions, the tracking performance of existing algorithms is often limited, especially in terms of unstable identity assignment and insufficient tracking accuracy. To address these challenges, this paper proposes a new multi-object tracking model—the Reparameterized and Global Context Track (RGTrack). This model is based on the Correlation-Sensitive Track (CSTrack) framework and innovatively introduces multi-branch training and attention mechanisms, combined with reparameterized convolutional networks and global attention modules, significantly enhancing the network’s feature extraction ability in complex scenes, especially in ignoring irrelevant information and focusing on key areas. It adopted a multiple association strategy to better establish the association relationship between targets in consecutive frames. Through this improvement, the Reparameterized and Global Context Track can better handle scenes with dense targets and severe occlusions, providing more accurate target identity matching and continuous tracking. Experimental results show that compared with the Correlation-Sensitive Track, the Reparameterized and Global Context Track has significant improvements in multiple key indicators: multi-object tracking accuracy (MOTA) increased by 1.15%, Identity F1 Score (IDF1) increased by 1.73%, and Mostly Tracked (MT) increased by 6.86%, while ID-switched (ID Sw) decreased by 47.49%. These results indicate that the Reparameterized and Global Context Track not only can stably track targets in more complex scenes but also significantly improves the continuity of target identities. Moreover, the Reparameterized and Global Context Track increased the frames per second (FPS) by 51.48% and reduced the model size by 3.08%, demonstrating its significant advantages in real-time performance and computational efficiency. Therefore, the Reparameterized and Global Context Track model maintains high accuracy while having stronger real-time processing capabilities, making it especially suitable for embedded devices and resource-constrained application environments.

## 1. Introduction

Multi-object tracking is one of the classical problems in computer vision engineering. It mainly involves a set of methods for accurately tracking the positions of multiple targets in a specific category in a video and giving each target different identity information [1]. Object detection usually inputs the image to be detected into the detection model. Finally, it uses rectangular boxes to display the position of interesting objects in the original image and determine their category. Unlike object detection, the MOT algorithm needs not only to output the location of the target of interest but also to associate the target’s identity with the location information output by the detection task and maintain the state of association between the identity information and the bounding box [2]. Most of today’s excellent multi-object tracking solutions use methods based on deep learning, among which the principal method is multiple-object Tracking By Detection (TBD) [3,4]. Specifically, this method identifies and detects targets in the video, and then the data association strategy is used to generate the target trajectory. Such methods finely decompose MOT into the following steps: object detection, feature extraction, motion prediction, similarity computation, and data association. The entire flow is illustrated in Figure 1. Firstly, the image is input into the detection model to obtain the position information of multiple objects in a single frame. Secondly, the tracking target is cut out to obtain the appearance information of the target. The target’s position in the next frame is obtained by motion prediction. Finally, the affinity matrix between the appearance information of the target in the previous frame and the position information of the target in the current frame is calculated in two adjacent frames. Each target’s unique assignment of identity information is obtained by maximum matching, and the current target is finally associated with the existing trajectory. The above process is recursive to complete the tracking of multiple targets.

According to the above analysis of TBD, we can find that this kind of method requires two computationally intensive components, the detection model and the embedding model, in which the embedding model often uses the person reidentification (ReID) model [5]. In recent years, many excellent detection methods and ReID models have emerged with the iterative updating of deep learning technology. Benefiting from the continuous updating and development of the two components, the accuracy of TBD methods is getting higher and higher. Although the TBD method performs well in tracking performance, it is challenging for it to meet the actual needs in terms of tracking speed. One of the possible reasons for its unsatisfactory speed is that most models need to process features through two computation-intensive components respectively. This leads to a long inference time of the whole system and finally leads to the problem of slow tracking speed with low real-time performance.

With the deepening of the research on MOT algorithms based on deep learning, a class of methods that integrate detection components and embedding models into a single model to complete the detection and appearance feature extraction tasks simultaneously has attracted much attention. The main reason is that this alleviates the speed bottleneck problem of multi-step multi-object tracking schemes. Inspired by You Only Look Once (YOLO) [6,7], a one-stage object detection framework in the field of object detection, which simultaneously completes the tasks of object classification and location, Wang et al. creatively proposed the JDE [8] paradigm based on YOLOv3 [9]. JDE converts multi-object tracking into multi-task learning. Three tasks of classification, regression, and appearance extraction are completed while a single network is performing forward propagation. By carefully designing the structure of the target detection network, the detection model and embedding model can share the same set of features, which avoids the repeated calculation of the detection and embedding steps, saves the network inference time, and improves the processing speed of the model. Finally, the experiment proves that the proposed method can run in near real-time speed, while the accuracy is comparable to that of the two-stage model.

Many researchers have recognized the multi-object tracking of the JDE paradigm since it was proposed. However, its application in actual scenes is still very challenging. In areas with high crowd density and heavy shading, the detection model has difficulty locating the object of interest, resulting in the detection box not being apparent, which, due to unreliable data correlation IDs, leads to the tracking effect not being ideal. The specific details of the JDE paradigm still need to be improved. CSTrack [10] is a representative method of improving the JDE model, which adds a cross-correlation network (CCN) and scale-aware attention network (SAAN) based on JDE, and alleviates the excessive competition problem of detection tasks and embedding tasks in JDE.

In order to improve the accuracy of multi-object tracking algorithms in occlusion scenes, this paper adds multi-branch training to the CSTrack framework, introduces the attention mechanism to deeply aggregate high-level semantics, and establishes an RGTrack model with enhanced detection ability.

The work in this paper can be summarized as follows:We introduce global context (GC) attention to enhance the detection ability of the network.We use the Reparameterized Visual Geometry Group (RepVGG) convolution module to add multi-branch training to improve the feature fusion ability of the network.We introduce the multiple association strategy to better establish the association relationship of the target between consecutive frames.

The remainder of this article is organized as follows. Section 2 reviews the related work on multi-object tracking. Section 3 provides a detailed description of the proposed RGTrack model. Section 4 presents the experimental results, and Section 5 discusses the findings and limitations. Finally, Section 6 concludes the paper and outlines directions for future research.

## 2. Related Work

According to whether the MOT model uses a single model for object detection and appearance extraction, we can divide the detection-based multi-object tracking research work into two categories: single models that perform object detection and appearance extraction separately, and single models that perform object detection and appearance extraction simultaneously.

### 2.1. SDE Model

The separate model transforms the video into image sequences to complete the specific tracking task through the detection model and the embedding model. Bewley et al. [11] used a simple combination of the Kalman filter and Hungarian algorithm in the tracking component and called it the SORT algorithm, using it to solve common problems in data association. They proposed a simple online tracking framework based on inter-frame prediction and association and used classical tracking methods to achieve advanced tracking quality. It achieved good performance in terms of speed and accuracy. Wojke et al. [12] analyzed the advantages and disadvantages of the SORT algorithm. In order to track long-term occluded targets and effectively reduce the number of identity switches, they proposed DeepSORT by combining appearance information with pre-trained association metrics. Compared with SORT, DeepSORT reduces the number of identity exchanges after occlusion disappearance by 45% and can also achieve good results at high resolution. Feichtenhofer et al. [13] proposed the D&T framework by summarizing the previous tracking methods. They established the ConvNet architecture of simultaneous detection and tracking, introduced relevant features to represent the co-occurrence of objects in time, and connected the cross-frame trajectory and frame-level detection, resulting in video-level high-precision detection. Experiments show that increasing the time stride can significantly improve the tracker’s speed. In order to associate unreliable detection results with existing trajectories, Chen et al. [14] adopted the detection strategy of collecting candidate detections from detection and tracking outputs to deal with unreliable detection. They proposed MOTDT, an online multi-person tracking framework based on the loss function of a fully convolutional neural network, then introduced the appearance representation of deep learning. Finally, training on a large-scale human recognition dataset improves the recognition ability when dealing with intra-class occlusion. In order to alleviate the problems of ReID, motion prediction, and occlusion processing, Bergmann et al. [15] proposed Tractor++. This universal tracker does not train and optimize the tracking data. They used the detector’s border regression to predict the target’s position in the next frame, convert the detector into a tracker, and deal with the simplest tracking scenarios.

### 2.2. JDE Model

Due to the decoupling of the object detection model and ReID model, the SDE method has tremendous advantages in multi-object tracking accuracy. However, its running speed is not sufficient to meet the actual demand. Integrating detection and appearance feature extraction with a single model is a timely emerging field. More and more scholars are committed to researching MOT algorithms for multiple tasks in a single model. Wang et al. [8] discussed the detection-based tracking paradigm, including the detection model of target positioning and the appearance embedding model of data association. They found that executing the two models separately would lead to efficiency problems because the running time was the sum of the two steps, and the potential sharing structure between them was not studied. Most of the existing MOT studies focus on the data association steps. In essence, it is a real-time association method rather than a real-time MOT system. Based on the above discussion and analysis, the JDE framework is proposed to realize the simultaneous learning of object detection and appearance embedding in a shared model. It dramatically reduces the running time of the whole MOT system and enables it to run at near real-time speed. Wang et al. [16] considered that the single-object detection in FairMOT only uses the attributes of the current frame but does not use the target information of the preceding and posterior frames. Hence, they proposed integrating the Graph Neural Network (GNN) into the object detection framework to form a joint framework. They used the GNN to extract relationships between objects and learn better features to improve detection and data association. Liang et al. [10] analyzed the joint process of object detection and ReID tasks in JDE. They found that the differences between the two tasks were ignored in the joint training process, which eventually caused performance problems. In order to alleviate the excessive competition between detection subtasks and reidentification subtasks in the MOT system, they proposed a mutual correlation network and scale-aware attention network and demonstrated the effectiveness of this method by ablation experiments.

## 3. Methods

With the deepening of the research on the one-stage detection algorithm YOLO, its performance in object detection has also steadily improved. Consistent with the CSTrack framework, this paper adopts the most widely used YOLOv5 [17] algorithm as the basic framework for research. Based on version 6.0, this paper studies the MOT algorithm for joint detection and embedding tasks in a single model. The overall structure of the network is shown in Figure 2, consisting of the feature extraction network backbone, feature fusion component neck, and network prediction component head. Within them, “Conv” is the abbreviation of “Convolutional Layer”, indicating a convolutional layer, which is one of the core modules used in deep learning for image processing. “C3” is the abbreviation of “Cross Stage Partial 3”, and it is a module design in the CSPNet (Cross Stage Partial Network) architecture. “GC” is the abbreviation of “Global Context Block”. “UpSample” is the abbreviation of “Upsampling Layer”, representing an upsampling layer. “Concat” is the abbreviation of “Concatenation Layer”, indicating a concatenation layer. “RepConv” is the abbreviation of “Reparameterized Convolution”. “CCN” is the abbreviation of “cross-correlation network”. “SAAN” is the abbreviation of “scale-aware attention network”.

### 3.1. Backbone

Feature extraction is performed to reduce the dimension of the original input data or recombine the original features for subsequent use, that is, to reduce the dimension of the data and to sort out the existing data features [18]. The feature extraction network of Yolo series algorithms is highly researchable, and much work has been performed to improve it [19,20]. In order to improve the detection ability of the MOT model, the feature reconstruction part of the feature extraction network is improved. Since multi-object tracking based on deep learning mainly adopts high-resolution images for training, the default input size of the network (640,640,3) is changed to (1088,608,3). Multiple convolutions are used to increase channel information and reduce the information loss caused by downsampling. Next, multiple convolutions with a convolution kernel of 3 × 3 and step of 2 are used to downsample the image while deepening the number of feature map channels. Then, high-level features are extracted through the global context module. Finally, the feature map relative to the original image at 1/4, 1/8, 1/16, and 1/32 is obtained. Due to the large resolution of the input feature map and considering the overall computational effort, the last three layers of the feature map are retained for subsequent operations such as feature fusion. Finally, the feature map is converted to a fixed-size feature vector using the SPPF layer.

#### Global Context Block

As convolutional networks suffer from low learning efficiency and cumulative over-deepness of the network, Wang et al. [21] proposed a generalized, simple, and non-local operation named Non-local. It can be embedded into the current network based on the non-local mean filtering operation in the field of picture filtering. They proposed self-attention [22], an attention mechanism for non-local information statistics based on capturing dependencies between long-range features. The general formula for Non-local is shown in Equation (1). Here, *x* represents the input signal, which is generally used in computer vision as a feature map representation, i represents the output location, f is a function that calculates the similarity of i to j, g is a representation of the feature map at location j, and C(x) represents the response factor.(1)$$y_{i}=\frac{1}{C(x)}\sum_{\forall j}f(x_{i},x_{j})g(x_{j})$$

Hu et al. [23] suggested correcting the features of channels by modeling the relationship between them and proposed implementing Senet Block by three operations, Squeeze, Excitation, and Reweight, to enhance the characterization ability of neural networks. Cao et al. [24] discussed the computational waste problem of non-local operators and the problem that SE Block cannot fully use global context information. Then they combined both advantages and proposed GC Block, which has the global context modeling capability of Non-local Block and can save computation like SE Block. Figure 3 shows that this module contains three operations: context modeling, channel transform, and feature fusion.

### 3.2. Neck

The feature extraction network [25] obtains visual, attribute, and semantic features in order and does not adequately understand the three hierarchical features. In this paper, we introduce Reparameterized Convolution (RepConv) in the neck fusion component to further process the backbone-extracted features and hierarchically fuse the three features to improve the detection model performance.

#### RepVGG Block

In classification networks, the backbone network is the benchmark network for many advanced tasks like object detection, and its performance largely determines part of the upper limit of that network, among which representative networks are AlexNet [26], ResNet [27], and VGGNet [28]. The key idea of the RepVGG [29] classification network is to use the multi-branching approach of ResNet to avoid gradient disappearance during training and to use the single-way mode of VGG during inference by operator fusion to ensure constant inference. Since RepVGG has the advantage of multi-branch training with constant inference speed, this paper replaces the convolutional layer with recombining parameter convolution. In the following, the advantages of the RepVGG module are described in detail in the training and inference phases. In the training stage, as shown in Figure 4, the original structure uses a 3 × 3 convolution kernel with a stride of 1 and a padding of 1 to downsample the feature map. RepConv has two more branches compared to the original convolution structure, one of which uses a 1 × 1 convolution kernel with a stride of 2 and a padding of 0 for standard convolution, and the other branch performs a batch normalization (BN) operation on the input map.

In the inference stage, as shown in Figure 5, the original 3 × 3 convolution with BN is kept unchanged first, after which the 1 × 1 convolution with BN and BN in the training branch are all converted to 3 × 3 convolutions with BN. Finally, the three 3 × 3 convolution kernels with BN are added separately to form one 3 × 3 convolution with BN operation.

The first problem in the inference phase is fusing the 3 × 3 convolution, 1 × 1 convolution, and no-convolution operations in the three branches into a single 3 × 3 convolution. Due to the linear characteristic of convolution, the original 3 × 3 convolution is kept unchanged, the 1 × 1 convolution is expanded to 3 × 3 convolution, and BN is expanded to a constant mapping so that the input and output of the constructed convolution layer are equal. As shown in Figure 6, the 1 × 1 convolution expansion specifically involves padding eight zeros around the 1 × 1 convolution kernel, and the padding is 1. As shown in Figure 7, the input channels are assumed to be 2, and the BN layer is expanded to have the center of the convolutional kernel corresponding to each channel of 1, and the rest are 0.

The second problem of the inference phase is how to fuse Conv and BN layers for inference acceleration. Considering that Conv and BN are linear operations, the Conv and BN modules can be set into a single linear operation for inference acceleration. For the convolutional layer, the number of channels in each convolutional kernel is the same as the number of channels in the input feature map. The number of convolutional kernels determines the number of channels in the output feature map. In contrast, the inference pattern BN layer (γ and β are known) contains four main parameters: mean μ, variance σ, and learnable parameters γ and β. Here μ and σ are the statistics of the training process, and γ and β are learned through the network. For the ith channel of the feature map, BN is calculated as in Equation (2). Therefore, the weight calculation formula of the ith convolution kernel of the new convolutional layer after transformation is as shown in Equations (3) and (4).(2)$$BN{\left(M,\mu ,\sigma ,\gamma ,\beta \right)}_{i}=(M_{i}-\mu_{i})\times \frac{\gamma_{i}}{\sigma_{i}}+\beta_{i}$$(3)$$W_{i}^{\prime }=\frac{\gamma_{i}}{\sigma_{i}}\times W_{i}$$(4)$$b_{i}^{\prime }=\beta_{i}-\frac{\mu_{i}\gamma_{i}}{\sigma_{i}}$$

After the above operation, introducing the RepConv module for multi-branch learning during training enables the network to focus on more helpful information. At the same time, the fusion of multiple branches into a single branch during inference avoids the problem of increasing the inference speed.

### 3.3. Head

In order to ensure that the detection task and the ReID task operate on the feature maps independently of each other, the three feature maps obtained by the feature fusion component are subjected to the CCN operation. Then the three different-sized process feature maps are input to the detection module and the SAAN module with scale awareness, respectively. The Detect module is used to perform the object detection task on the feature map, predict the anchor boxes by the grid, and perform the classification loss and regression loss calculation based on the class and location of the ground truth. The SAAN module fuses three different scales of different channel feature maps to unify the ReID feature scale and then captures the relevant information by convolution according to the total number of IDs to output a 1D vector representation of the corresponding feature map.

#### 3.3.1. CCN Operation

The core objective of the CCN is to enhance feature representation capability, aiming to extract general and specific features that are more suitable for detection and ReID tasks. It achieves this by exploring the autocorrelation between feature channels and the cross-correlation between tasks, thereby improving the model’s ability to capture key features and obtaining more robust and discriminative feature representations in different tasks.

The input is the feature map $F\in R^{C\times H\times W}$ from the neck part. Here, *C* represents the number of channels, and *H* and *W* denote the height and width of the feature map. To reduce computational complexity while retaining key statistical information, an average pooling operation is first performed on the feature map to obtain the reduced-dimensional feature map $F^{\prime }\in R^{C\times H\prime \times W\prime }$. Here, *H′* < *H* and *W′* < *W*. Next, two different convolutional layers are used to process the reduced-dimensional feature map, generating two intermediate feature maps ***T*_1_** and ***T*_2_**. These two feature maps correspond to the feature representation branches for the detection task and the ReID task, respectively. Subsequently, ***T*_1_** and ***T*_2_** are reshaped into matrix forms ***M*_1_** and ***M*_2_**, with dimensions *C* × *N*′, where *N*′ = *H*′ × *W*′.

To capture the channel dependencies within each task, the CCN computes the self-attention graph. Specifically, $W_{T_{1}}=softmax(M_{1}\cdot M_{1}^{T})$ represents the self-attention of the detection branch, reflecting the correlation between the relevant channels for detection; similarly, $W_{T_{2}}=softmax(M_{2}\cdot M_{2}^{T})$ represents the self-attention of the ReID branch, emphasizing the relationship between the identity discrimination channels. To facilitate knowledge sharing and complementarity between tasks, the CCN also computes the cross-attention graph. $W_{S_{1}}=softmax(M_{1}\cdot M_{2}^{T})$ represents the cross-task attention from the ReID branch to the detection branch, while $W_{S_{2}}=softmax(M_{2}\cdot M_{1}^{T})$ represents the attention from detection to ReID. These cross-attention graphs help the model learn the shared semantic information between the two tasks.

Subsequently, the self-attention and cross-attention are fused. For the detection branch, the fused attention map is $W_{1}=W_{T_{1}}+W_{S_{1}}$. For the ReID branch, it is $W_{2}=W_{T_{2}}+W_{S_{2}}$. This fusion mechanism retains the task-specific feature responses while introducing cross-task context information. Finally, the weighted and reconstructed original feature maps are obtained using the fused attention map. Specifically, the output feature maps $F_{T_{1}}=W_{1}\cdot F+F$ and $F_{T_{2}}=W_{2}\cdot F+F$ are achieved, achieving feature enhancement. Through the residual connection structure, the original information is not damaged, and the response of the key channels is enhanced.

In summary, the CCN achieves effective enhancement of the features for detection and ReID tasks through steps such as dimensionality reduction, branch processing, self-attention calculation, cross-attention calculation, attention fusion, and feature reconstruction. This structure not only improves the performance of a single task but also promotes collaborative learning among multiple tasks.

#### 3.3.2. Detection Loss Calculation

The Detect module splits the input feature map into grids. It then maps the grid coordinates to the original map, with each grid predicting three metrics: rectangular box, confidence, and classification probability [30]. The rectangular box characterizes the size and position of the object. The confidence characterizes the confidence level of the predicted rectangular box and ranges from 0 to 1. The classification probability characterizes the class of the target. The loss function measures the distance between predicted and expected information. The closer the predicted information is to the expected information, the smaller the value of the loss function. Training contains three main aspects of loss: rectangular box loss, confidence loss, and classification loss. Thus, the loss function of detection is defined as in Equation (5).(5)$$Loss_{d}=\alpha \times Loss_{box}+\beta \times Loss_{obj}+\gamma \times Loss_{cls}$$

The overall loss is the weighted sum of three losses. Usually *β* is 0.4, and *α* and *γ* are both equal to 0.3. In this paper, we use the siou loss [31] to calculate the rectangular box loss, and both confidence and classification losses are calculated by cross-entropy loss.

#### 3.3.3. Appearance Embedding Loss Calculation

Appearance embedding requires that the distance metric of different targets should be large enough, so this network treats the appearance embedding problem as a classification problem. The embedding dimension is mapped into categories of the total number of IDs by a linear classification layer. Finally, the loss is calculated according to the cross-entropy loss of the output and the label. Suppose an anchor box instance in a batch of samples is $f^{T}$, a positive sample is $f^{+}$, and a negative sample is $f^{-}$. In the loss calculation, we focus on all negative sample classifications. Here, $f^{T}f^{-}$ denotes the probability that the anchor instance is considered a positive sample and $f^{T}f_{j}^{-}$ denotes the probability that the anchor is considered the *j*th category, where the subscript *i* denotes the *i*th sample. As in Equation (6), the appearance embedding loss is calculated using a form similar to the cross-entropy loss.(6)$$Loss_{e}=\frac{1}{N}\sum_{i=1}^{N}-\log\frac{e^{f_{i}^{T}f_{i}^{+}}}{e^{f_{i}^{T}f_{i}^{+}}+\sum_{j}e^{f_{i}^{T}f_{i,j}^{-}}}$$

As in Equation (7), the total loss is the weighted sum of the two losses. Here α and β are task-independent uncertainties, which are parameters learned based on the data distribution of the dataset.(7)$$Loss_{total}=\alpha \times Loss_{d}+\beta \times Loss_{e}$$

### 3.4. Multiple Association Strategies

Based on YOLOv5, a data association module was integrated to establish the association relationship of targets between consecutive frames. Specifically, we used Kalman filtering to track the detected targets, thereby achieving multi-target tracking. The description of the entire association process is shown in the following Figure 8.

For the first frame of the tracking sequence, the status of all detection boxes is recorded and they are added to the historical trajectory set, marking them as in the “pending” state. If there are detection boxes that match the trajectory in the next three consecutive frames, then the “pending” state of the trajectory is converted to the “tracked” state.

When conducting subsequent tracking of the sequence, there are four matching processes: first association, where the high-score detections that failed to match are re-matched with the joint trajectories; second association, where the low-score detections and the trajectories in the tracking state are matched; third association, where the high-score detections are matched with the undetermined trajectories; and fourth association, where all the trajectories are matched. Before the matching, based on the confidence score of the detection box, the detections are classified as high-score detections and low-score detections. At the same time, the Kalman filter is used to predict the positions of the trajectories in the tracking state and the undetermined state. Then, the distance measurement matrix EG is calculated, and its calculation method can be obtained from Equation (8).(8)$$\begin{array}{c} {EG}_{i,j}=Embedding\left(f\left(i\right),f\left(j\right)\right)+\frac{GIOU\left(i,KF\left(j\right)\right)}{2} \end{array}$$

In Equation (8), f represents the appearance information, *Embedding* is a function that measures the cosine distance of the appearance information, *KF* is the Kalman filtering operation, and *GIOU* is a function that measures the intersection-over-union ratio between the detection box and the motion prediction box. Compared with the original IOU distance measurement, the value range of *GIOU* measurement has been doubled. Therefore, a scaling factor is added in front of it to reduce the contribution of *GIOU* measurement to the total distance.

The first matching process is to match the high-score detections with the associated trajectories, resulting in unmatched trajectories, unmatched high-score detections, and matched detections and trajectories. For the matched high-score detections, they are added to the corresponding trajectories and the trajectories are updated; for the unmatched joint trajectories, their states will be judged and the temporarily stored trajectories and the tracked trajectories will be separated. If the temporarily stored state exceeds the set maximum trajectory storage duration, the trajectory will be deleted; otherwise, the temporarily stored state will be retained until the next frame; for the unmatched high-score detections, their information will be retained.

During the second matching process, the high-score detection that failed in the first match and the trajectories that failed the match and are in the tracking state will be matched using IOU. The same three results will be obtained. If the high-score box successfully matches the tracking state trajectory, the trajectory will be updated; for the high-score detection that failed the tracking, it will be retained for the fourth match; for the tracking state trajectory that failed the match, it will be retained for the third match with the low-score detection box.

During the third matching process, the low-score detection bounding boxes are matched with the trajectories of the tracking states that failed in both matching attempts. Three results are obtained. For the trajectories of the low-score detections that were successfully matched with the tracking states, the trajectory states are updated; for the unmatched low-score detections, they are deleted; and for the unmatched tracking states of the detections, their states are changed to a pending state.

During the fourth matching process, the unmatched high-score detections retained from the second matching are matched with the undetermined trajectories using IOU (intersection over union) to obtain three results. For the high-score detections that are successfully matched, they are added to the assigned trajectory and the trajectory is updated; for the unmatched high-score detections, it is considered that a new target has entered the frame, a new trajectory is created for them, and they are marked as undetermined; for the undetermined trajectories that are not successfully matched, they are deleted.

## 4. Experimental Results and Analysis

### 4.1. Datasets and Evaluation Criteria

#### 4.1.1. Datasets

In line with the training set setup of CSTrack, the training set used in this paper is a mixture of six publicly available datasets for pedestrian detection and multi-object tracking. It contains the ETH dataset [32], the CityPersons dataset [33], the CalTech dataset [34], the CUHK-SYSU dataset [35], the PRW dataset [36], and the MOT17 dataset [37], where the MOT17 video image sequence data are identical to MOT16, and the only difference between the two is that the MOT17 annotation is more accurate and diverse. Similar to the JDE validation set, the training process used the CalTech dataset for the ReID task and the CityPersons dataset for the detection task. Finally, the MOT16 test set with no annotation information was evaluated and submitted to the MOT server for tracking effect evaluation. The number of images, the number of annotated boxes, and the number of IDs for the mixed dataset are listed in Table 1. After data processing, the training set included 53,580 images, 275,000 bounding boxes, and 15547 ID annotations.

#### 4.1.2. Measurement

Multi-object tracking is a task that includes object detection and data association, and it is difficult to evaluate the performance of a multi-object tracking system with a single metric. The metric evaluation of this system should include the following three characteristics: timeliness, consistency, and stability. Specifically, all appearing objects should be able to be found in time. The object position should match the actual object position as much as possible. Each object should be assigned a unique ID, and this ID assigned to the object should remain constant throughout the sequence. In this paper, we use the CLEAR MOT metric from the MOT Challenge online evaluation system, which is generally accepted in academia, to evaluate the performance of the multi-object tracking system proposed. The main evaluation metrics and the meaning of each metric are shown in Table 2, where “↑” values in parentheses indicate that the higher the value, the better the performance, and “↓” values in brackets indicate that the lower the value, the better the performance.

This paper selects the recognized MOTA and IDF1 indicators as the primary evaluation indicators. MOTA includes the number of times a tracker performs wrong object positioning and matching, and IDF1 includes whether a tracker can track an object for a long time. The formulations of MOTA and IDF1 are shown in Equations (9) and (10), where GT denotes the number of annotated object boxes, IDTP denotes the number of object boxes with correctly assigned IDs, and IDFP denotes the number of object boxes with incorrectly assigned IDs. IDFN denotes the number of object boxes that are not assigned IDs in the annotation.(9)$$MOTA=1-\frac{FN+FP+IDSw}{GT}$$(10)$$IDF1=\frac{2}{\frac{1}{IDP}+\frac{1}{IDR}}=\frac{2IDTP}{2IDTP+IDFP+IDFN}$$

### 4.2. Experimental Parameter Setting

The experimental hardware setup, including an Intel Core i9-11900K CPU, PM9A1 NVMe Samsung 1024 GB SSD, NVIDIA GeForce RTX 3090 GPU, and 128 GB memory, was purchased in Changsha, China. The main software environment is Ubuntu 18.04 LTS, Pycharm2022.1.1, python3.8, torch1.8.1, and cuda toolkit 11.3. The improved version 6.0 is taken as the overall network of multi-object tracking tasks. In order to accelerate the convergence speed of the network, its pre-training weight on the COCO dataset is loaded, and the silu activation function is adopted. The predefined anchor boxes on the three feature maps are as follows: [8,24, 11,34, 16,48], [32,96, 45,135, 64,192], and [128,384, 180,540, 256,640]. In the training process, the SGD optimization algorithm is used for 100 epochs of training. The decay factor is 5 × 10^−4^, the initial learning rate is 5 × 10^−3^, the final learning rate is 5 × 10^−4^, the batch size is 16, and the embedding dimension is 512. In addition, data enhancement techniques such as multi-scale input, random cropping, color dithering, and image flipping are used to prevent model overfitting. In the testing phase, since the MOT challenge limits the number of submissions and the test set does not contain annotation information, the trained weights are used to fine-tune the model by experimentally verifying the effect on MOT16-train with annotation information. The fine-tuned model experiments on MOT16-test are conducted, and finally, the obtained detection information and identity information are stored in text documents and submitted to the server for evaluation.

### 4.3. Experimental Results

Table 3 shows the results of the proposed model RGTrack compared with the JDE improved algorithm CSTrack on the MOT16 training set. In general, RGTrack obtained a 1.70% improvement in MOTA and a 1.22% improvement in IDF1. From the results of the model predictions, RGTrack is less likely to predict false negative and false positive cases than the CSTrack model. Therefore, MOTA gained a boost, indicating that the addition of GC attention to the feature extraction network achieved some effect. From the results of the data association, RGTrack reduced the total number of ID Sw to 308. Therefore, the IDF1 metric was also improved, indicating that introducing multi-branch training in the feature fusion is practical. Compared with CSTrack, the model size of RGTrack has been reduced by 3.08%. It can effectively reduce the complexity of the model and is suitable for deployment on small mobile devices.

Figure 9 shows the training curve results. Figure 9a depicts the error variation in the model’s regression of target bounding boxes during the training process. From the graph, it can be seen that this loss value was relatively high in the initial stage of training, approximately 0.045, indicating that the model’s ability to locate the target position was weak at the beginning. As the training progressed, the curve rapidly decreased and stabilized after about 50 epochs, ultimately converging to approximately 0.015. This trend indicates that the model gradually learned more precise positioning features and could effectively reduce the deviation between the predicted box and the real box. The continuous decrease in the loss without significant fluctuations also indicates that the training process is relatively stable, and the optimization strategies (such as learning rate setting and gradient update) are reasonable, without experiencing severe oscillations or divergence.

Figure 9b shows the confidence loss of the model in judging the existence of the target on the training set. At the beginning of the training, this value was approximately 0.035, indicating that the model’s ability to distinguish the target area was limited at the initial stage, and it was prone to mistakenly identifying the background as the target or failing to detect the real target. As the training progressed, the curve continued to decline and converged around 50 epochs, ultimately stabilizing at approximately 0.018. This downward trend indicates that the model gradually enhanced its sensitivity to the target area, was able to more accurately distinguish the target from the background, reflecting the stability of the training process, without signs of overfitting or training collapse, and improved the reliability of the detection.

Figure 9c measures the proportion of actual positive samples among the predicted positive samples by the model, which represents the “accuracy” of the detection results. In the early stage of training, the precision rate was relatively low, close to 0.2, indicating that the model generated a large number of false positives. However, as training progressed, the precision rate rose rapidly, reaching above 0.8 after approximately 20 epochs, and remained at a high level in subsequent training. This significant improvement indicates that as the train/box_loss and train/obj_loss curves decrease, the model not only improves its localization ability but also significantly reduces false positive phenomena, and the output detection results are more reliable. The high and stable precision rate means that the model has strong discriminative ability and practicality under the current validation conditions.

Table 4 shows the ablation experiments of the module RepVGG GC Block proposed in this paper on CSTrack. The dataset for this ablation experiment is set as follows: the first half of MOT17-train is used for the training set, MOT15-val is used for validation during the training process, and the second half of MOT17-train is finally used for model performance qualification. For subsequent comparison, the models in Table 4 are named A, B, C, and D in that order. In terms of run speed FPS, models B and D with Rep did not show a significant reduction in run speed compared to their counterparts A and C without the module (27.92 vs. 28.09, 27.52 vs. 27.56). Comparing A with B, A performs well in IDF1 and ID Sw (69.7% vs. 68.4%, 442 vs. 519). This indicates that the addition of the reconfiguration parameter convolution can better fuse features at different levels during the feature fusion phase, improving the association ability of the network. Comparing A with C, we can find that C is slightly weaker than A in predicting negative samples (16,353 vs. 16,340), and its running speed decreases from 28.09 to 27.56. At the same time, C outperforms A in all other aspects, which indicates that the C model loses some speed with the addition of the global context module. However, it can better focus on important information in the feature extraction phase and enhance the model’s detection capability. Comparing A with D, we can find that D combines the advantages of both B and C. Apart from a slight decrease in running speed (27.52 vs. 28.09), its tracking performance (61.9% vs. 61.0%), association effect (70.2% vs. 68.4%), and detection performance (3908 vs. 4151, 15,777 vs. 16,340) are all improved. This indicates that adding the reconfiguration parameter convolution module and the global context module can improve tracking performance with a slight loss of processing speed.

Table 5 compares the results of the different methods on the MOT16 test set. This paper divides the networks in the table into two categories: two-stage and one-stage methods. The methods with “*” in the table are one-stage methods, and the rest are two-stage methods. Since FPSs are different with different test hardware, the running speed results in this paper are all on the hardware and software environment described in Chapter 3. The two-stage approach will have a lower FPS than the current value as the corresponding FPS term only corresponds to the correlation part of the data, and in practice, the fusion detection step will take longer. The inference speed of the one-stage methods is related to all steps in the system, from detection to association. From Table 5, we can conclude that the JDE inference speed is much higher than that of the particular detection embedding method with comparable tracking accuracy. However, its IDF1 scores do not reach the same level, which shows that the JDE model does not effectively handle frequent occlusion cases. Compared with JDE’s improved algorithm CSTrack, the method proposed in this paper not only maintains the real-time operation speed but also increases the MOTA by 1.15% and the IDF1-related accuracy by 1.73%. Moreover, the MT of this model has increased by 6.86%, and ID Sw has decreased by 47.49%. This indicates that this model can more accurately track more targets and better maintain the consistency of target identities when dealing with complex scenarios. The FPS of this model has increased by 51.48%, and the model size has decreased by 3.08%. This demonstrates a significant improvement in the detection speed of the model, making it suitable for embedded devices. Furthermore, the proposed RGTrack model outperforms multiple existing detection models in terms of detection accuracy, detection speed, and model lightweighting. This demonstrates the novelty of the proposed model.

### 4.4. Visualization Results

To visually demonstrate the superiority of the RGTrack model, Figure 10 shows the detection results of the CSTrack and the improved RGTrack models for detecting dense crowds. The left figures (a) and (c) represent the inference results of the CSTrack model, while the right figures (b) and (d) represent the inference results of the improved model RGTrack. The green boxes indicate the correct detections made by the model, the red boxes represent the missed detections, and the blue boxes are the incorrect predictions. By comparing Figure 10a with Figure 10b, it can be seen that the CSTrack model missed detecting the blue pedestrians in the scene, while the improved model detected all the targets in the scene. By comparing Figure 10c with Figure 10d, it can be found that the CSTrack model had five red boxes and three blue boxes (i.e., five missed detections and three false detections) compared to the original annotations, while the RGTrack model had five missed detections and one false detection. Overall, the RGTrack model has a certain improvement effect in detection accuracy compared to the CSTrack model.

Figure 11 shows the tracking results of CSTrack and RGTrack. The left image is the tracking result of CSTrack, and the right image is the tracking result of RGTrack. At frame 177, CSTrack detected 52 pedestrians, while RGTrack detected 59 pedestrians. In the central area of the image, due to the limited ability to detect occluded pedestrians, CSTrack identified the three pedestrians at the bottom of the picture as two targets and assigned two IDs (6, 70), while RGTrack accurately detected three pedestrians and assigned three IDs (36, 67, 78). Among the 10 pedestrians on the right side of the image, CSTrack detected 8 pedestrians, but failed to detect the occluded black-shirted and gray-shirted pedestrians, while RGTrack detected all 10 pedestrians and assigned IDs 56 and 72 to the black-shirted and gray-shirted pedestrians, respectively. Additionally, in the densely populated area of the upper left corner of the image, CSTrack detected 15 pedestrians, while RGTrack detected 19 pedestrians. At frame 204, CSTrack detected 55 pedestrians, while RGTrack detected 59 pedestrians. CSTrack still has the problem of poor detection in dense areas, only detecting one target (ID: 31) for the three pedestrians carrying bags, while RGTrack tracked the pedestrians that were severely occluded with IDs 104 and 107, and accurately tracked the three pedestrians carrying bags (ID: 6, 31, 34). At frame 495, CSTrack detected 67 pedestrians, while RGTrack detected 70 pedestrians. In the upper right corner of the image, CSTrack changed the ID of the stationary green-shirted pedestrian from 44 to 202, while RGTrack continuously tracked it, and the IDs assigned to the pink-shirted and two white-shirted pedestrians were 92, 134, and 177. Overall, the number of IDs assigned in CSTrack is excessive, resulting in inaccurate tracking. The tracking effect of RGTrack is better than that of CSTTrack, and it can generate more stable trajectories.

## 5. Discussion

Inspired by the representative model of the JDE framework, we proposed a multi-object tracking algorithm to solve the problem that the inaccurate detection ability of the JDE model in occlusion scenarios leads to unstable ID allocation in data association and unsatisfactory tracking accuracy. By introducing the global attention mechanism in the feature extraction stage and using multi-branch training in the fusion stage, the detection task and appearance embedding task are well integrated with one model. The experimental results show that the model proposed in this paper can achieve a good tracking effect with a slight loss of inference speed and a small number of parameters added, which provides support for implementing the algorithm in intelligent video surveillance.

However, this study also has limitations. For the proposed new method, although it has achieved comprehensive improvements in detection accuracy, detection speed, and model size, there is still room for further enhancement to better achieve a balance between model complexity and detection accuracy. The specific challenges are as follows:(1)In complex scenarios, the target may be obscured by other objects or other targets, which can lead to tracking loss or confusion of identity. The existing tracking algorithms have difficulty maintaining the correct identity of the target under prolonged occlusion.(2)Multi-target tracking often requires processing a large number of targets in a large-scale scenario, which leads to high computational complexity and time consumption. Especially when the hardware resources are limited, how to balance the tracking accuracy and computational efficiency remains a challenge.(3)When the target approaches or intersects, an identity switch (ID switch) may occur, meaning that the tracker misidentifies the target’s identity. Although there are some methods to reduce ID switches, this remains a challenge in scenarios with high-density targets.(4)Different application scenarios, lighting conditions, background complexities, and target appearances and behavior patterns present various challenges for multi-target tracking. The generalization ability of existing models can still be enhanced.

To address the aforementioned issues, in future research directions, integrating deep learning with multimodal data (such as RGB images, depth maps, LiDAR data, etc.) can provide more comprehensive environmental and target information, thereby improving tracking accuracy and robustness. Future research can focus on how to effectively conduct long-term tracking and reidentification when the target reappears, especially in cases where the target has been out of contact for a long time. Real-time online learning and adaptive models can dynamically adjust the tracking strategy based on the changes in the target (such as appearance changes or motion changes) to enhance the accuracy and stability of tracking. Graph optimization methods can better model the spatial and temporal relationships between targets and reduce false associations and ID switches. Future research may further delve into this direction. To meet the real-time requirements of multi-target tracking in practical applications, how to improve the algorithm’s computational efficiency without sacrificing tracking accuracy is an important research direction for the future. Utilizing hardware acceleration technologies such as GPUs and TPUs may play a significant role in real-time tracking applications.

## 6. Conclusions

In this paper, we have presented a novel multi-object tracking framework, the Reparameterized and Global Context Track (RGTrack), to address the limitations of existing joint detection and embedding (JDE) models in complex, occluded, and crowded scenes. By building upon the Correlation-Sensitive Track (CSTrack) architecture, RGTrack introduces a global attention mechanism and multi-branch training strategy to enhance feature representation, enabling more accurate detection and robust appearance embedding under challenging conditions. The integration of reparameterized convolutional blocks further strengthens the model’s capacity while maintaining efficiency. Additionally, a multiple association strategy is employed to improve cross-frame data association, significantly reducing identity switches. Experimental results demonstrate that RGTrack achieves substantial improvements over the baseline: MOTA increases by 1.15%, IDF1 by 1.73%, and MT by 6.86%, while ID switches are reduced by 47.49%. Notably, the model also achieves a 51.48% increase in FPS and a 3.08% reduction in model size, highlighting its superior balance between accuracy and efficiency. These advantages make RGTrack particularly suitable for real-time applications on embedded and resource-constrained platforms. While challenges remain in long-term occlusion, high-density interactions, and generalization across diverse environments, our work provides a strong foundation for practical multi-object tracking. Future efforts will explore multimodal fusion, online adaptation, and graph-based optimization to further advance tracking robustness and scalability.

Multi-target tracking also has many potential applications. In autonomous driving systems, it is necessary to track various targets such as vehicles, pedestrians, traffic signs, etc., on the road in real time. Efficient multi-target tracking can provide reliable dynamic environmental information for decision-making systems. In security surveillance systems, MOT can be used to track multiple suspicious individuals or objects and promptly trigger alarms, enhancing public safety. In sports events, MOT can be used to analyze the movements of athletes, team tactics, and key events during the competition. In navigation systems of robots or drones, MOT can help them identify and track dynamic obstacles or targets around them, achieving efficient path planning and obstacle avoidance. In medical image analysis, MOT can be used to track the dynamic changes in cells, organs, or tissues, providing support for disease diagnosis and treatment.

## Figures and Tables

**Figure 1 sensors-25-05325-f001:**
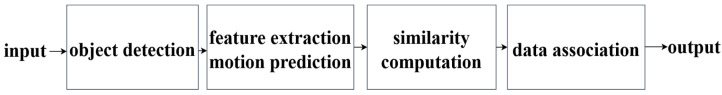
Process diagram of multi-object tracking based on detection.

**Figure 2 sensors-25-05325-f002:**
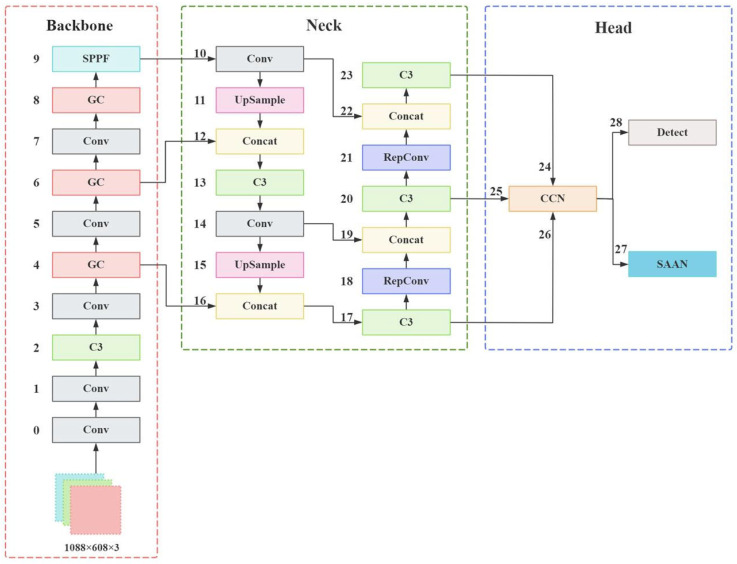
The overall network structure of RGTrack.

**Figure 3 sensors-25-05325-f003:**
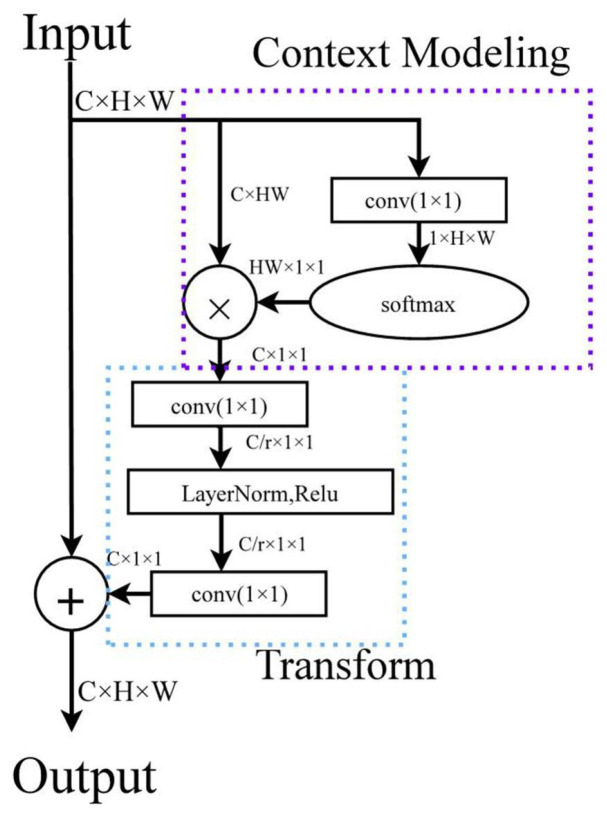
The operational flow of global context attention.

**Figure 4 sensors-25-05325-f004:**
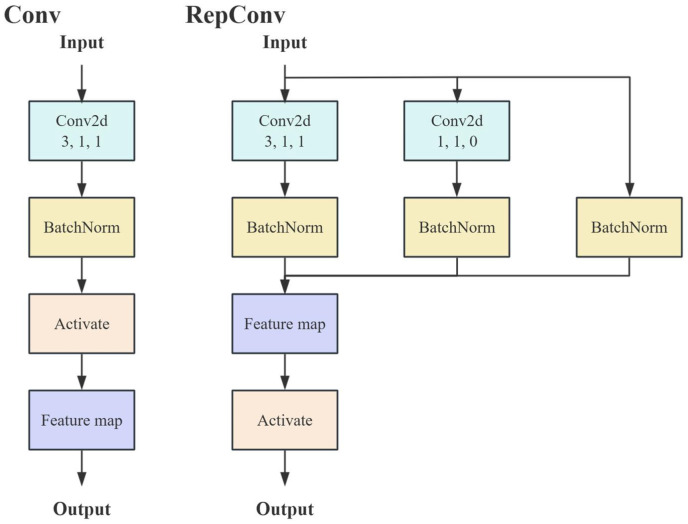
Step comparison of Conv and RepConv.

**Figure 5 sensors-25-05325-f005:**
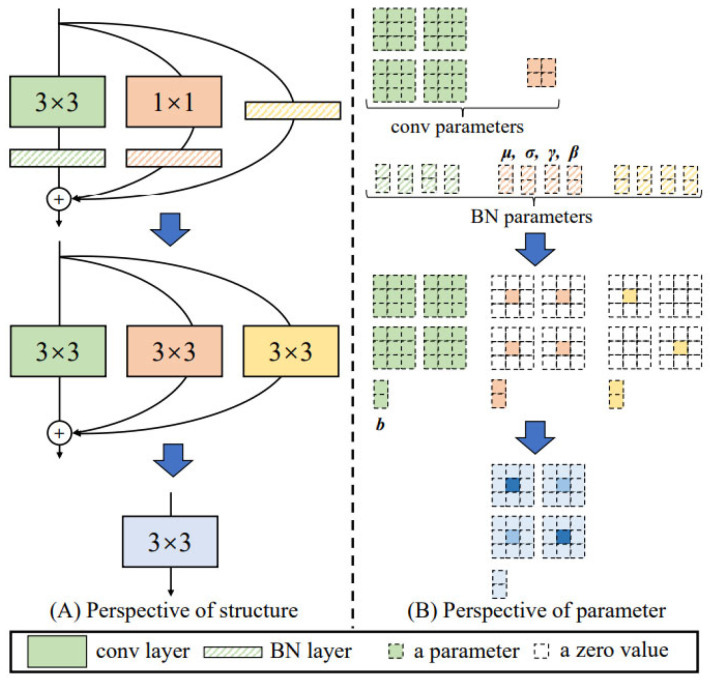
Flow diagram of reorganization parameters in the inference phase.

**Figure 6 sensors-25-05325-f006:**
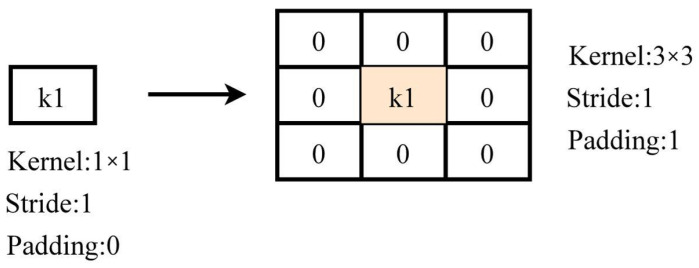
Diagram of the 1 × 1 convolution conversion to the 3 × 3 convolution.

**Figure 7 sensors-25-05325-f007:**
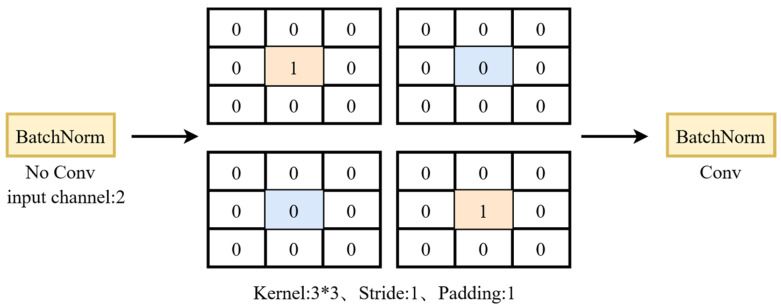
Diagram of the BN layer conversion to a 3 × 3 convolution.

**Figure 8 sensors-25-05325-f008:**
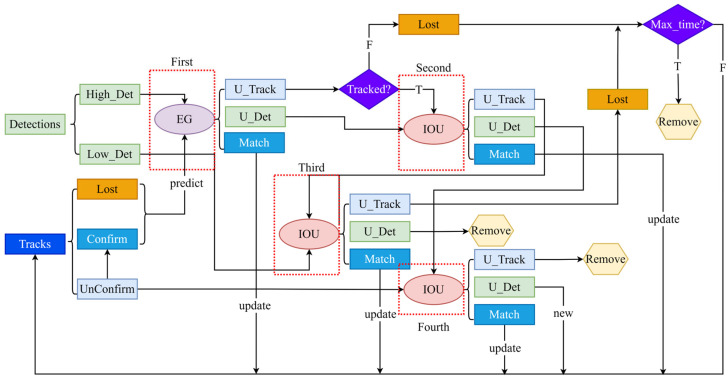
Process diagram of the multiple association strategy.

**Figure 9 sensors-25-05325-f009:**
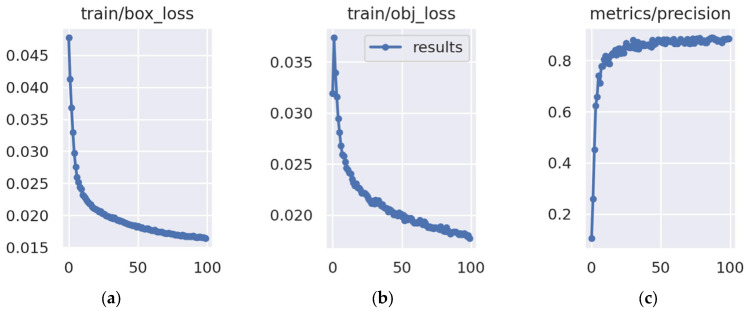
Training curve results. (**a**) The bounding box regression loss on the training set; (**b**) object confidence loss on the training set; (**c**) the precision rate of the model.

**Figure 10 sensors-25-05325-f010:**
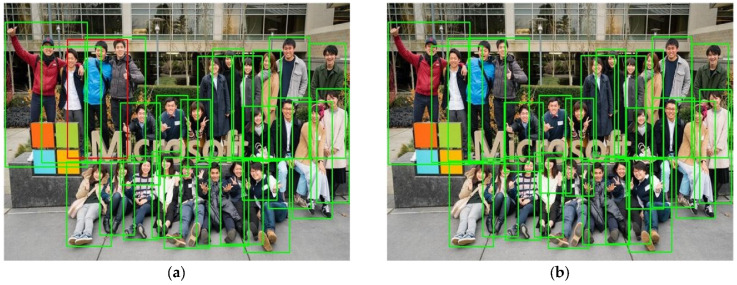
Demonstration of detection effects of CSTrack and RGTrack. (**a**,**c**) The detection effect of CSTrack; (**b**,**d**) the detection effect of RGTrack.

**Figure 11 sensors-25-05325-f011:**
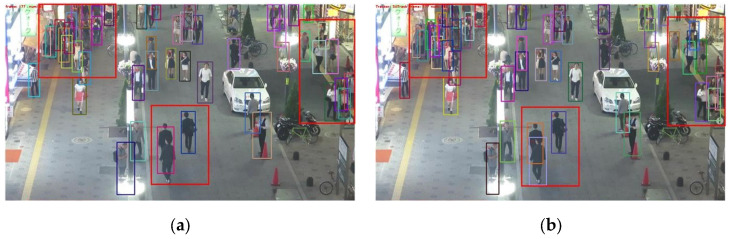
The tracking results of CSTrack and RGTrack. (**a**) The CSTrack detection result for frame 177; (**b**) the RGTrack detection result for frame 177; (**c**) the CSTrack detection result for frame 204; (**d**) the RGTrack detection result for frame 204; (**e**) the CSTrack detection result for frame 495; (**f**) the RGTrack detection result for frame 495.

**Table 1 sensors-25-05325-t001:** Summary of training datasets [9].

Dataset	ETH	CP	CT	MOT17	CS	PRW	Total
Imgs	2 K	3 K	27 K	5 K	11 K	6 K	54 K
Boxes	17 K	27 K	46 K	112 K	55 K	18 K	275 K
Identities	-	-	1 K	1.6 K	12 K	0.9 K	15.5 K

**Table 2 sensors-25-05325-t002:** CLEAR MOT metrics for evaluation of multi-object tracking.

Measurement	Description
MOTA (↑)	Multiple-object tracking accuracy, a standard evaluation index for multiple-object tracking, is the first index for comprehensive evaluation of trackers.
IDF1 (↑)	The Identification F1 Score of ID is the first index to judge the matching performance of a tracker.
MT (↑)	A maximum tracking target number is the number of tracks for which the target tracking trajectory accounts for more than 80% of the annotated track length.
FP (↓)	False positive is the total number of incorrectly detected objects in the video sequence.
FN (↓)	False negative is the video sequence’s total number of missed detection targets.
ID Sw (↓)	Identification switch is the total number of times a trajectory has been ID-switched.

**Table 3 sensors-25-05325-t003:** Comparison of RGTrack and CSTrack results on MOT16-train.

Method	MOTA (%)	IDF1 (%)	FP	FN	ID Sw	Model Size (MB)
CSTrack [10]	82.4	82.1	5126	13,798	545	48.7
RGTrack	83.8	83.1	4971	12,652	308	47.2

**Table 4 sensors-25-05325-t004:** Ablation experiment of RGTrack in MOT17-train.

Method	Rep	GC	MOTA (%)	IDF1 (%)	ID Sw	FP	FN	FPS
CSTrack (A)			61.0	68.4	519	4151	16,340	**28.09**
R-CSTrack (B)	√		60.2	69.7	442	4254	16,774	27.92
G-CSTrack (C)		√	61.2	68.9	504	4055	16,353	27.56
RGTrack (D)	√	√	61.9	70.2	384	3908	15,777	27.52

**Table 5 sensors-25-05325-t005:** Comparison of results of different methods on MOT16 test set.

Method	MOTA (%)	IDF1 (%)	MT (%)	ID Sw	FPS	Model Size (MB)
DeepSORT-2 [12]	61.4	62.2	32.8	781	8.1	137.13
CNNMTT [38]	65.2	62.2	32.4	946	6.4	85.8
POI [39]	66.1	65.1	34.0	805	6	94.1
Tube_TK [40]	63.0	58.6	32.1	1137	3.0	155.1
GSDT [16]	66.2	68.7	31.5	1318	4.9	144.2
CTrackerV1 [41]	66.6	57.4	31.2	1529	6.8	139.4
CJTracker * [42]	58.7	58.2	35.5	877	16.0	95.1
QuasiDense * [43]	68.7	66.3	35.6	1378	17.3	69.7
JDE * [9]	64.4	55.8	35.4	1544	18.8	45.8
CSTrack * [10]	69.4	69.3	35.0	958	16.9	48.7
RGTrack *	70.2	70.5	37.4	503	25.6	47.2

* The method marked with “*” belongs to a single-stage approach.

## Data Availability

The data used in this study are publicly available.

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
