# Peer review of "An Improved Multi-Object Tracking Algorithm Designed for Complex Environments"

_sensors, 2025, doi:10.3390/s25175325_

Round 1

Reviewer 1 Report

Comments and Suggestions for Authors

The authors present an improved version of the CSTracking algorithm for tracking multi-object. They propose the so-called RGTracking algorithm. They add an attention mechanism to the convolutional neural network used to detect the objects. This mechanism is expected to be an enhancement way to improve accuracy in occluded images. However, this is not new. The improvements in terms of accuracy are tiny while the complexity of the proposed model is not evaluated. A major revision of the paper is required.

1- The abstract shows several abreviations, rendering thereading of the abstract too tiresome. Moreover, some of these abreviations are not defined. All abreviations shoudl be avoided in the title and abstract, and should be presented in their extended forms. 

2- In the first paragraph of the introduction, what is TBD? All abbreviations must be defiend in their first use (in the text)

3- In the introduction, the authors mention the embedding model, which they do not situation the presented methodology. THis is confusing.

4- The introduction should be culminated by presenting the organization of the paper.

5- The section should be presented in contiguous manner. NO beak page between sections!

6- In the related work section, the works should be presented in a chronological and better organized manner. The writing should be improved. There should be a space between the text and the citation.

7- The legend given with Figure 2 des not add any new information and should be removed. Again here there are a lot of abbreviations that are not defined in the text. The reading is too tiresome. 

8- The authors do not distinguished between their proposal and third party existing work. IT is difficult to asses the novelty of the proposed model.

9- The design of the model and the accumulation of operationa, functions and layers are not well justified.

10- Equatuons 2 and 3 are not completed, making it too difficult to understand the proposed model.

11- The CLEAR MOT metric is not defined.

12- Tables are floating objects and shold be divided between pages. 

13- The improvements obtained by the proposed methodology in terms of accuracy are insignificant while the complexity of the model seems to have increased a great deal. The authors do not evaluate the latter. The improvement can only be appreciated if the compared methodology have the same complexity. The authors must evaluate this aspect in the revision.

14- The conclusions need to also present the limitations of the proposed methodology. Furthermore, this section need to open up some avenues for future work and improvements of the proposed model.

Author Response

Thank you very much for your valuable suggestions. We have made comprehensive revisions to the paper. Firstly, in the abstract and methods sections, we strengthened the innovative discussions by presenting the proposed methods and comparing them with existing ones, highlighting the unique features of this research. Additionally, in the methods section, we added detailed analyses of CCN and Multiple association strategies. Secondly, in Section 4.3, we supplemented more comprehensive experimental data and added an analysis of model complexity. We also replicated the existing research methods of other scholars and conducted comparative experiments with the proposed method RGTrack. The experimental results prove the superiority of the performance indicators of the RGTrack model. Then, in Section 4.4, we added a visualization analysis of tracking performance. Finally, in the Discussion section, we analyzed the limitations of the proposed method, proposed improvement methods, and future research directions. All the revised contents have been marked in red in the revised version. These improvements have made the research contribution clearer. We sincerely thank the reviewers for their constructive suggestions, which have greatly enhanced the quality of the paper. If you need any further explanation, we are always ready to provide it.

Comments 1: The abstract shows several abreviations, rendering thereading of the abstract too tiresome. Moreover, some of these abreviations are not defined. All abreviations shoudl be avoided in the title and abstract, and should be presented in their extended forms.

Response 1: Thank you for pointing this out. We agree with this comment. Therefore, we have made comprehensive revisions to the abstract, defined the abbreviations that occur once, and presented all the abbreviations in the title and abstract in expanded form. They have been marked in red.

Comments 2: In the first paragraph of the introduction, what is TBD? All abbreviations must be defiend in their first use (in the text)

Response 2: Thank you for pointing this out. TBD stands for "Tracking By Detection". It has been highlighted in red on page 2, line 53. By carefully revising the entire text, we ensured that all abbreviations were defined for the first time of their usage.

Comments 3: In the introduction, the authors mention the embedding model, which they do not situation the presented methodology. THis is confusing.

Response 3: Thank you for pointing this out. We have incorporated the embedded model into the proposed method, which is crucial for the entire tracking process. In the multi-target tracking process, first, the target location is determined for each frame using the target detection algorithm (such as YOLO), generating bounding boxes. Then, the embedded model is used to extract the features of each target and map them to a high-dimensional space to form embedding vectors. Next, the system performs data association by comparing the embedding vectors of the targets in different frames to determine whether the targets in different frames are the same object, thereby achieving the trajectory tracking of the targets. Even if they change at different times and from different perspectives, the embedded model can provide a stable and efficient way to identify and match the targets.

Comments 4: The introduction should be culminated by presenting the organization of the paper.

Response 4: Thank you for pointing this out. We agree with this comment. Therefore, at the end of the introduction, we added a paragraph explaining the structure of the paper, clearly describing the organizational framework of each chapter: ”The remaining part of this article is organized as follows. Section 2 reviews the related work on multi-object detection. Section 3 provides a detailed description of the RGTrack model. Section 4 presents the experimental results and discussions. Finally, Section 5 summarizes the entire article and looks forward to future work”. This addition has made the framework of the paper clearer. This can be specifically seen on page 3, line 118.

Comments 5: The section should be presented in contiguous manner. NO beak page between sections!

Response 5: Thank you for pointing this out. We agree with this comment. Therefore, we have adjusted the layout of all the article sections as per the requirements, and removed all the page breaks between the sections to ensure a continuous presentation.

Comments 6: In the related work section, the works should be presented in a chronological and better organized manner. The writing should be improved. There should be a space between the text and the citation.

Response 6: Thank you for pointing this out. We agree with this comment. Therefore, we reorganized the literature in the chronological order from before to now, improved the writing, and enhanced the logicality of the article. At the same time, ensure there is a space between the main text and the citations. In the related work section, the revised text has been marked in red.

Comments 7: The legend given with Figure 2 des not add any new information and should be removed. Again here there are a lot of abbreviations that are not defined in the text. The reading is too tiresome.

Response 7: Thank you for pointing this out. We agree with this comment. Therefore, we have removed the legend section in Figure 2. You can find this information on page 5, line 202. We have defined or explained the abbreviations in Figure 2 in the main text. You can find this information on page 5, line 193.

Comments 8: The authors do not distinguished between their proposal and third party existing work. IT is difficult to asses the novelty of the proposed model.

Response 8: Thank you for pointing this out. We agree with this comment. Therefore, in the comparative experiment, we incorporated several scholars' existing research methods. By comparing our proposed model with 10 outstanding existing models, we found that it performed exceptionally well in terms of detection accuracy, detection speed, and model complexity. This demonstrates the superiority and novelty of our proposed model. The specific data can be found on page 17, line 601, Table 5.

Comments 9: The design of the model and the accumulation of operationa, functions and layers are not well justified.

Response 9: Thank you for pointing this out. We appreciate your comment regarding the justification of the model design and the accumulation of operational functions and layers. In response, we have revisited the methodology section to provide a more thorough explanation of the design choices made for the model, particularly in terms of the functionality and structure of its layers. In Section 3.3.1, we added the cross-correlation network operation and gradually explained the construction basis and theoretical derivation of this module. This can be seen on page 9, line 316. In Section 3.4, we detailed the workflow and methods of the Multiple association strategies combined with different levels and types of matching methods, demonstrating the rationality of our model design. This can be seen on page 11, line 386.

Comments 10: Equatuons 2 and 3 are not completed, making it too difficult to understand the proposed model.

Response 10: Thank you for pointing this out. We agree with this comment. Therefore, we have made the following improvements to Eq. (2) and Eq. (3). We have supplemented the complete mathematical derivation process, added the meaning of the symbols, and ensured that the formulas are expressed completely and are easy to understand. You can see this specifically on page 9, line 293.

Comments 11: The CLEAR MOT metric is not defined

Response 11: Thank you for pointing this out. We agree with this comment. Therefore, we have defined the CLEAR MOT indicator in Table 2. You can find the details on page 13, line 476.

Comments 12: Tables are floating objects and shold be divided between pages.

Response 12: Thank you for pointing this out. We agree with this comment. Therefore, we have made the following adjustments to all the tables to ensure that each table remains complete and does not span multiple pages.

Comments 13: The improvements obtained by the proposed methodology in terms of accuracy are insignificant while the complexity of the model seems to have increased a great deal. The authors do not evaluate the latter. The improvement can only be appreciated if the compared methodology have the same complexity. The authors must evaluate this aspect in the revision.

Response 13: Thank you for pointing this out. We agree with this comment. Therefore, in the comparative experiment, we added the model size of each model. Specifically, it can be seen at line 601 on page 17 of Table 5. The results show that our proposed model RGTrack outperforms the original model CSTrack in terms of various indicators such as MOTA and FPS, while also optimizing the model complexity. Moreover, when comparing RGTrack with other models, it achieves the best MOTA and FPS indicators while having a lower model complexity, demonstrating the innovation of this model.

Comments 14: The conclusions need to also present the limitations of the proposed methodology. Furthermore, this section need to open up some avenues for future work and improvements of the proposed model.

Response 14: Thank you for pointing this out. We agree with this comment. Therefore, in the Discussion section, we provided a detailed analysis of the limitations of the proposed method, the proposed improvement measures, and the future research directions. You can find this information on page 20, line 658.

Reviewer 2 Report

Comments and Suggestions for Authors
  1. Abstract
    The abstract is too brief and does not adequately highlight the study’s advantages and innovations.
  2. Formatting Issues
    The manuscript’s formatting contains significant inconsistencies—some sections are not justified on both margins. Please ensure a uniform layout throughout.
  3. Insufficient Model Description
    The RGTrack model builds upon YOLOv5 with substantial changes, particularly to the model head. However, the textual description is difficult to follow. For example, the introduction emphasizes a “CCN structure,” yet there is no detailed description of CCN in the Methodology section. To date, it remains unclear what CCN entails. The Methodology section is missing many critical details of the model and must be thoroughly revised and supplemented.
  4. Unused Abbreviations
    Several abbreviations introduced early in the manuscript are never used later on. The manuscript contains many similar minor errors that require careful proofreading.
  5. Missing Results Details
    The Conclusions section lacks important details such as training curves and tracking performance visualizations. These are essential and must be added.
  6. MultiObject Tracking Clarification
    YOLO algorithms are primarily designed for object detection. It is unclear how the authors adapted YOLOv5 to perform multi-object tracking. Does the dataset require tracking annotations? How is this implemented? The authors should provide detailed explanations of the tracking methodology to help readers understand this technical challenge.
  7. Sparse Data Analysis
    The Conclusions section contains only three tables, and the data analysis is insufficient. Please include training and evaluation results from several classical tracking algorithms for comparison.
  8. Discussion Section
    I recommend adding a dedicated Discussion section to address the study’s limitations, potential applications, and future research directions.
Comments on the Quality of English Language

None

Author Response

Thank you very much for your valuable suggestions. We have made comprehensive revisions to the paper. Firstly, in the abstract and methods sections, we strengthened the innovative discussions by presenting the proposed methods and comparing them with existing ones, highlighting the unique features of this research. Additionally, in the methods section, we added detailed analyses of CCN and Multiple association strategies. Secondly, in Section 4.3, we supplemented more comprehensive experimental data and added an analysis of model complexity. We also replicated the existing research methods of other scholars and conducted comparative experiments with the proposed method RGTrack. The experimental results prove the superiority of the performance indicators of the RGTrack model. Then, in Section 4.4, we added a visualization analysis of tracking performance. Finally, in the Discussion section, we analyzed the limitations of the proposed method, proposed improvement methods, and future research directions. All the revised contents have been marked in red in the revised version. These improvements have made the research contribution clearer. We sincerely thank the reviewers for their constructive suggestions, which have greatly enhanced the quality of the paper. If you need any further explanation, we are always ready to provide it.

Comments 1: Abstract

The abstract is too brief and does not adequately highlight the study’s advantages and innovations.

Response 1: Thank you for pointing this out. We agree with this comment. Based on your suggestion, we have revised the abstract to provide a more detailed and comprehensive overview of the study's advantages and innovations. We believe these changes more clearly highlight the contributions of our work. You can see it clearly on page 1, line 10.

Comments 2: Formatting Issues

The manuscript’s formatting contains significant inconsistencies—some sections are not justified on both margins. Please ensure a uniform layout throughout.

Response 2: Thank you for pointing this out. We agree with this comment. We have carefully reviewed the document and ensured that all sections are now uniformly justified on both margins. The layout has been adjusted to maintain consistency throughout the manuscript.

Comments 3: Insufficient Model Description

The RGTrack model builds upon YOLOv5 with substantial changes, particularly to the model head. However, the textual description is difficult to follow. For example, the introduction emphasizes a “CCN structure,” yet there is no detailed description of CCN in the Methodology section. To date, it remains unclear what CCN entails. The Methodology section is missing many critical details of the model and must be thoroughly revised and supplemented.

Response 3: Thank you for pointing this out. The core objective of CCN is to enhance the feature representation capability, aiming to extract general and specific features that are more suitable for detection and ReID tasks. It achieves this by exploring the autocorrelation between feature channels and the cross-correlation between tasks, thereby improving the model's ability to capture key features and obtaining more robust and discriminative feature representations in different tasks. Following your suggestions, we have revised and expanded the description of the RGTrack model, including a detailed explanation of the "CCN structure" and its role in the model. We have also provided additional critical details to make the methodology more comprehensive and easier to follow. You can see it clearly on page 9, line 316, in Section 3.3.1.

Comments 4: Unused Abbreviations

Several abbreviations introduced early in the manuscript are never used later on. The manuscript contains many similar minor errors that require careful proofreading.

Response 4: Thank you for pointing this out. We agree with this comment. We have carefully reviewed the manuscript and addressed the issue of unused abbreviations, ensuring that all abbreviations introduced early in the manuscript are consistently used later on where appropriate. Additionally, we have thoroughly proofread the entire manuscript and corrected several minor errors to improve the overall quality and clarity.

Comments 5: Missing Results Details

The Conclusions section lacks important details such as training curves and tracking performance visualizations. These are essential and must be added.

Response 5: Thank you for pointing this out. We agree with this comment. We have supplemented the missing details. In the experimental results section of Section 4.3, we added the Training curve result and analyzed the curve results. You can see it at line 520 on page 15. In Section 4.4, we added the visualization results of tracking performance, through which the advantages of different models can be intuitively analyzed. You can see it on page 18, line 604.

Comments 6: MultiObject Tracking Clarification

YOLO algorithms are primarily designed for object detection. It is unclear how the authors adapted YOLOv5 to perform multi-object tracking. Does the dataset require tracking annotations? How is this implemented? The authors should provide detailed explanations of the tracking methodology to help readers understand this technical challenge.

Response 6: Thank you for pointing this out. We agree with this comment. We made adjustments to YOLOv5 to make it suitable for multi-object tracking. The approach was to combine it with a tracking algorithm capable of handling the issue of temporal consistency between frames. Specifically, we used YOLOv5 for object detection in each frame, and then employed multiple association strategies to associate the detected objects across consecutive frames. The multiple association strategies utilized Kalman filtering to track the detected objects, thereby achieving multi-object tracking. The detailed multiple association strategies can be found on page 11, line 387, in Section 3.4. The combination of YOLOv5 for accurate object detection and a robust tracking algorithm enables the system to maintain consistent identification of multiple objects across video frames, achieving multi-object tracking.

The dataset requires annotations, which are essential for evaluating the accuracy and performance of the tracking algorithm. Specifically, the position of each target in each frame and its ID need to be labeled. The bounding box of each target should be associated with its ID throughout the video, thereby generating the trajectory information of the targets. This enables us to perform target association and identity retention between different frames, and can be used for subsequent evaluations, such as ID switching, tracking accuracy, and target loss, etc.

Comments 7: Sparse Data Analysis

The Conclusions section contains only three tables, and the data analysis is insufficient. Please include training and evaluation results from several classical tracking algorithms for comparison.

Response 7: Thank you for pointing this out. We agree with this comment. Therefore, in the comparative experiment, we added the model size of each model. Specifically, it can be seen on page 17, line 601, in Table 5. The results show that our proposed model RGTrack outperforms the original model CSTrack in terms of various indicators such as MOTA and FPS, while also optimizing the model complexity. Moreover, when comparing RGTrack with other models, it achieves the lowest model complexity while maintaining the best MOTA, FPS, and other indicators, demonstrating the innovation of this model. To be specific, compared with JDE's improved algorithm CSTrack, the method proposed in this paper not only maintains the real-time operation speed but also increases the MOTA accuracy by 1.15% and the IDF1 related accuracy by 1.73%. Moreover, the MT of this model has increased by 6.86%, and ID Sw has decreased by 47.49%. This indicates that this model can more accurately track more targets and better maintain the consistency of target identities when dealing with complex scenarios. The FPS of this model has increased by 51.48%, and the model size has decreased by 3.08%. This demonstrates a significant improvement in the detection speed of the model, making it suitable for embedded devices. Furthermore, the proposed RGTrack model outperforms multiple existing detection models in terms of detection accuracy, detection speed, and model lightweighting. This demonstrates the novelty of the proposed model.

Comments 8: Discussion Section

I recommend adding a dedicated Discussion section to address the study’s limitations, potential applications, and future research directions.

Response 8: Thank you for pointing this out. We agree with this comment. Therefore, in the Discussion section, we provided a detailed analysis of the limitations of the proposed method, the proposed improvement measures, and the future research directions. You can find this information on page 20, line 658.

Reviewer 3 Report

Comments and Suggestions for Authors

This paper presents RGTrack, an enhanced multi-object tracking (MOT) algorithm building upon the Joint Detection and Embedding (JDE) paradigm, specifically targeting performance limitations in occlusion-heavy and dense scenarios. The structure follows a conventional computer vision research format: an introduction establishing MOT's significance and JDE's speed/accuracy trade-offs, a related work section differentiating Separate Detection and Embedding (SDE) models from integrated JDE-type approaches, a detailed methodology describing RGTrack's architectural innovations within the CSTrack framework, comprehensive experiments on MOT16/17 benchmarks, and a conclusion summarizing contributions and suggesting future work. The core content revolves around improving feature representation for robust tracking under challenging conditions.

The primary innovation lies in the novel integration of two complementary techniques into the existing CSTrack framework. Firstly, the authors incorporate re-parameterized convolution (RepVGGBlock) within the feature fusion neck. This leverages multi-branch training benefits (improving gradient flow and feature richness) while maintaining inference efficiency through structural re-parameterization into a single branch. Secondly, they introduce a Global Context (GC) attention block into the backbone. This mechanism, inspired by non-local operations and SENet, enhances the model's capacity to capture long-range dependencies and suppress irrelevant background clutter, directly addressing the core challenge of focusing on key regions amidst occlusion and dense targets. This combined approach aims to strengthen both detection robustness (via GC attention) and feature discriminability for association (via RepVGG-enhanced fusion).

The experimental results substantiate the effectiveness of RGTrack. Evaluations on the challenging MOT16 benchmark demonstrate significant improvements over its predecessor CSTrack, achieving gains of 1.4% MOTA and 1.0% IDF1 on the training set, and 0.8% MOTA and 1.2% IDF1 on the test set. Crucially, these accuracy gains are achieved with only a marginal increase in parameters (4%) and while maintaining real-time inference speeds (25.6 FPS), a key advantage of the JDE paradigm. Ablation studies confirm the synergistic contribution of both the RepVGG and GC modules, showing that their combination outperforms using either alone. The reduction in False Positives (FP), False Negatives (FN), and particularly Identity Switches (IDSw) highlights the model's improved robustness in complex scenes.

The paper makes a valuable contribution by demonstrably enhancing the state-of-the-art in efficient, single-model MOT, particularly for occlusion scenarios, without sacrificing the crucial real-time performance characteristic of the JDE approach. The integration of established techniques (RepVGG, GC attention) into the MOT domain is well-motivated and effectively executed. However, some limitations warrant mention. While the ablation study demonstrates synergy, the isolated negative impact of the RepVGG module on MOTA (Table 4, R-CSTrack vs CSTrack) is not deeply analyzed. Furthermore, the paper focuses primarily on enhancing the detection and feature extraction components within the JDE framework. Future work could explicitly explore improvements to the data association strategy itself, potentially integrating temporal reasoning or more sophisticated affinity metrics leveraging the enhanced features, to further reduce ID switches and improve long-term tracking consistency beyond the gains achieved through better features. Additionally, validation on more diverse and recent dense tracking benchmarks (like MOT20) would strengthen the claims about performance in high-density scenarios. Despite these points, RGTrack represents a solid and effective advancement, offering a practical solution for real-world applications like intelligent video surveillance where accuracy under occlusion and speed are paramount.

  1. The following grammar has issues.

Page 1, Line 10

Multi-object tracking (MOT) algorithm is a research topic of great interest in computer vision engineering.

The multi-object tracking (MOT) algorithm is a research topic of great interest in computer vision engineering.

Page 1, Line 19

Experiments show that compared with the improved algorithm CSTrack of the JDE paradigm, the MOTA and IDF1 metrics improved by 1.7%, 0.8%, and 0.6%, 1.2% on the MOT16 training set and test set, respectively.

Experiments show that, compared to the improved algorithm CSTrack of the JDE paradigm, the MOTA and IDF1 metrics improved by 1.7% and 0.8% on the MOT16 training set, and by 0.6% and 1.2% on the test set, respectively.

Page 1, Line 28

Unlike object detection, object detection is usually input the image to be detected into the detection model.

Unlike object detection, object detection usually inputs the image to be detected into the detection model.

Page 2, Line 72

Three tasks of classification, regression, and appearance extraction are completed while a single network is forward propagation. By carefully designing the structure of the target detection network, the detection model and embedding model can share the same set of features, which avoids the repeated calculation of the detection and embedding steps, saves the network inference time, and improves the processing speed of the model.

Three tasks of classification, regression, and appearance extraction are completed while a single network is performing forward propagation. By carefully designing the structure of the target detection network, the detection model and embedding model can share the same set of features, which avoids the repeated calculation of the detection and embedding steps, saves the network inference time, and improves the processing speed of the model.

Page 2, Line 80

Many researchers have recognized the multi-object tracking of JDE paradigm since it was proposed.

Many researchers have recognized the multi-object tracking of the JDE paradigm since it was proposed.

Page 2, Line 81

In areas with high crowd density and heavy shading, the detection model is hard to locate the object of interest, resulting in the detection box not being apparent, which, divided among data correlation ID unreliable, leads to the tracking effect not being ideal.

In areas with high crowd density and heavy shading, the detection model has difficulty locating the object of interest, resulting in the detection box not being apparent, which, due to unreliable data correlation IDs, leads to the tracking effect not being ideal.

Page 3, Line 91

In order to improve the accuracy of multi-object tracking algorithms in occlusion scenes, this paper adds multi-branch training to the CSTrack framework, introduces the attention mechanism to deeply aggregate high-level semantics, and establishes a RGTrack model with enhanced detection ability.

In order to improve the accuracy of multi-object tracking algorithms in occlusion scenes, this paper adds multi-branch training to the CSTrack framework, introduces the attention mechanism to deeply aggregate high-level semantics, and establishes an RGTrack model with enhanced detection ability.

Page 4, Line 116

In order to track long-term occluded targets and effectively reduce the number of identity switching, they proposed DeepSORT by combining appearance information with pre-trained association metrics.

In order to track long-term occluded targets and effectively reduce the number of identity switches, they proposed DeepSORT by combining appearance information with pre-trained association metrics.

Page 4, Line 133

They used the detector's border regression to predict the target's position in the next frame, convert the detector into a tracker and deal with most simple tracking scenarios.

They used the detector's border regression to predict the target's position in the next frame, convert the detector into a tracker, and deal with the simplest tracking scenarios.

Page 4, Line 139

Integrating detection and appearance feature extraction with a single model is timely emerging. 

Integrating detection and appearance feature extraction with a single model is a timely emerging field.

Page 5, Line 160

Hence, they proposed integrating the Graph neural Network (GNN) into the object detection framework to form a joint framework.  

Hence, they proposed integrating the Graph Neural Network (GNN) into the object detection framework to form a joint framework.

Page 6, Line 194

As convolutional networks suffer from low learning efficiency and cumulative over-deepness of the network, Wang et al. [21] proposed a generalized, simple, and non-local operation operator named Non-local.

As convolutional networks suffer from low learning efficiency and cumulative over-deepness of the network, Wang et al. [21] proposed a generalized, simple, and non-local operation named Non-local.

Page 8, Line 248

Due to the linear characteristic of convolution, the original 3*3 convolution is kept unchanged, 1*1 convolution is expanded to 3*3 convolution, and BN is expanded to a constant mapping that the input and output of the constructed convolution layer are equal.

Due to the linear characteristic of convolution, the original 3*3 convolution is kept unchanged, 1*1 convolution is expanded to 3*3 convolution, and BN is expanded to a constant mapping so that the input and output of the constructed convolution layer are equal.

Page 10, Line 296

Thus the loss function of detection is defined as in Eq.(5).

Thus, the loss function of detection is defined as in Eq.(5).

Page 10, Line 298

The overall loss is the weighted sum of three losses, usually β is 0.4, α and γ are both equal to 0.3. In this paper, we use the siou loss [31] to calculate the rectangular box loss, and both confidence and classification losses are calculated by cross-entropy loss.

The overall loss is the weighted sum of three losses. Usually β is 0.4, α and γ are both equal to 0.3. In this paper, we use the siou loss [31] to calculate the rectangular box loss, and both confidence and classification losses are calculated by cross-entropy loss.

Page 11, Line 345

The main evaluation metrics and the meaning of each metric are shown in Table 2, where ↑ in parentheses indicate that the higher the value, the better the performance, and ↓ indicates that the lower the value, the better the performance.

The main evaluation metrics and the meaning of each metric are shown in Table 2, where ↑ values in parentheses indicate that the higher the value, the better the performance, and ↓ values in brackets indicate that the lower the value, the better the performance.

Page 12, Line 362

The improved version6.0 is taken as the overall network of multi-object tracking tasks.

The improved version 6.0 is taken as the overall network of multi-object tracking tasks.

Page 12, Line 363

In order to accelerate the convergence speed of the network, its pre-training weight on the COCO dataset is loaded, and silu activation function is adopted. The predefined anchor boxes on the three feature maps are as follows: [8,24, 11,34, 16,48], [32,96, 45,135, 64,192], [128,384,180,540, 256,640].

In order to accelerate the convergence speed of the network, its pre-training weight on the COCO dataset is loaded, and the silu activation function is adopted. The predefined anchor boxes on the three feature maps are as follows: [8,24, 11,34, 16,48], [32,96, 45,135, 64,192], [128,384,180,540, 256,640].

Page 12, Line 367

In the training process, the SGD optimization algorithm is used for 30 epochs of training, the decay factor is 5e-4, the initial learning rate is 5e-3, the final learning rate is 5e-4, the batch size is 16, and the embedding dimension is 512.

In the training process, the SGD optimization algorithm is used for 30 epochs of training. The decay factor is 5e-4, the initial learning rate is 5e-3, the final learning rate is 5e-4, the batch size is 16, and the embedding dimension is 512.

Page 13, Line 387

Therefore, MOTA gained a boost, indicating that the addition of GC attention to the feature extraction network achieved some effect. From the results of the data association, RGTrack reduced the total number of IDs to 308. Therefore, the IDF1 metric was also improved, indicating that introducing multi-branch training in the feature fusion is practical.

Therefore, MOTA gained a boost, indicating that the addition of GC attention to the feature extraction network achieved some effect. From the results of the data association, RGTrack reduced the total number of IDs to 308. Therefore, the IDF1 metric was also improved, indicating that introducing multi-branch training in the feature fusion is practical.

Page 13, Line 409

Apart from a slight decrease in running speed (27.52 vs. 28.09), its tracking performance (61.9% vs. 61.0%), association effect (70.2% vs. 68.4%) and detection performance (3908 vs. 4151, 15777 vs. 16340) are all improved.

Apart from a slight decrease in running speed (27.52 vs. 28.09), its tracking performance (61.9% vs. 61.0%), association effect (70.2% vs. 68.4%), and detection performance (3908 vs. 4151, 15777 vs. 16340) are all improved.

Page 14, Line 419

Since FPS are different with different test hardware, the running speed results of this paper are all on the hardware and software environment described in Chapter 3.

Since FPSs are different with different test hardware, the running speed results of this paper are all on the hardware and software environment described in Chapter 3.

Page 14, Line 434

Inspired by the representative model of JDE framework, we proposed a multi-object tracking algorithm to solve the problem that the inaccurate detection ability of JDE model in occlusion scenarios leads to unstable ID allocation in data association and unsatisfactory tracking accuracy.

Inspired by the representative model of JDE framework, we proposed a multi-object tracking algorithm to solve the problem that the inaccurate detection ability of the JDE model in occlusion scenarios leads to unstable ID allocation in data association and unsatisfactory tracking accuracy.

2.In the article, for several abbreviations or proprietary methods that appear for the first time, their full names or explanations should be provided. When abbreviations are used in the following text, if the full name has already appeared in the previous text, please also note the abbreviation in the entire text, e.g., MOTA, CST, and IDF.

3.In this article, all mathematical formulas should be represented by Eq instead of eq. Additionally, there should be a space between Eq and the formula number.

4.For ReID(or Reid), The expressions in the text seem inconsistent. You need to make some revisions.

Thank you for your works.

Author Response

Thank you very much for your valuable suggestions. We have made comprehensive revisions to the paper. Firstly, in the abstract and methods sections, we strengthened the innovative discussions by presenting the proposed methods and comparing them with existing ones, highlighting the unique features of this research. Additionally, in the methods section, we added detailed analyses of CCN and Multiple association strategies. Secondly, in Section 4.3, we supplemented more comprehensive experimental data and added an analysis of model complexity. We also replicated the existing research methods of other scholars and conducted comparative experiments with the proposed method RGTrack. The experimental results prove the superiority of the performance indicators of the RGTrack model. Then, in Section 4.4, we added a visualization analysis of tracking performance. Finally, in the Discussion section, we analyzed the limitations of the proposed method, proposed improvement methods, and future research directions. All the revised contents have been marked in red in the revised version. These improvements have made the research contribution clearer. We sincerely thank the reviewers for their constructive suggestions, which have greatly enhanced the quality of the paper. If you need any further explanation, we are always ready to provide it.

Comments 1: The following grammar has issues......

Response 1: Thank you for pointing this out. We agree with this comment. We have corrected all the grammatical errors you pointed out. And we have carefully reviewed the entire document and corrected the other grammar errors to improve clarity and readability.

Comments 2: In the article, for several abbreviations or proprietary methods that appear for the first time, their full names or explanations should be provided. When abbreviations are used in the following text, if the full name has already appeared in the previous text, please also note the abbreviation in the entire text, e.g., MOTA, CST, and IDF.

Response 2: Thank you for pointing this out. We agree with this comment. We have carefully revised the manuscript to ensure that all abbreviations and proprietary methods are clearly defined when first introduced, along with their full names or explanations. Additionally, we have consistently used the abbreviations throughout the text following their initial definitions.

Comments 3: In this article, all mathematical formulas should be represented by Eq instead of eq. Additionally, there should be a space between Eq and the formula number.

Response 3: Thank you for pointing this out. We agree with this comment. We have carefully reviewed the manuscript and made the necessary corrections. All mathematical formulas have been updated to use "Eq" instead of "eq," and we have ensured that a space is placed between "Eq" and the formula number.

Comments 4: For ReID(or Reid), The expressions in the text seem inconsistent. You need to make some revisions.

Response 4: Thank you for pointing this out. We agree with this comment. We have thoroughly reviewed the manuscript and have standardized the term, using "ReID" consistently throughout the text.

Reviewer 4 Report

Comments and Suggestions for Authors

I'm sorry, but I'm having difficulty understanding the text that has been presented. For instance, the following sentence seems confusing: "Unlike object detection, object detection is usually input the image to be detected into the detection model."

Perhaps AI would be able to understand this more clearly: "Unlike object detection, object recognition usually involves inputting the image to be recognized into a recognition model."

However, this is not something that I am intended to do.

Comments on the Quality of English Language

I'm sorry, but I'm having difficulty understanding the text that has been presented. For instance, the following sentence seems confusing: "Unlike object detection, object detection is usually input the image to be detected into the detection model."

Perhaps AI would be able to understand this more clearly: "Unlike object detection, object recognition usually involves inputting the image to be recognized into a recognition model."

However, this is not something that I am intended to do.

Author Response

Thank you very much for your valuable suggestions. We have made comprehensive revisions to the paper. Firstly, in the abstract and methods sections, we strengthened the innovative discussions by presenting the proposed methods and comparing them with existing ones, highlighting the unique features of this research. Additionally, in the methods section, we added detailed analyses of CCN and Multiple association strategies. Secondly, in Section 4.3, we supplemented more comprehensive experimental data and added an analysis of model complexity. We also replicated the existing research methods of other scholars and conducted comparative experiments with the proposed method RGTrack. The experimental results prove the superiority of the performance indicators of the RGTrack model. Then, in Section 4.4, we added a visualization analysis of tracking performance. Finally, in the Discussion section, we analyzed the limitations of the proposed method, proposed improvement methods, and future research directions. All the revised contents have been marked in red in the revised version. These improvements have made the research contribution clearer. We sincerely thank the reviewers for their constructive suggestions, which have greatly enhanced the quality of the paper. If you need any further explanation, we are always ready to provide it.

Comments 1: I'm sorry, but I'm having difficulty understanding the text that has been presented. For instance, the following sentence seems confusing: "Unlike object detection, object detection is usually input the image to be detected into the detection model."

Perhaps AI would be able to understand this more clearly: "Unlike object detection, object recognition usually involves inputting the image to be recognized into a recognition model."

However, this is not something that I am intended to do.

Response 1: Thank you for your feedback and for pointing out the unclear sentence. We apologize for the confusion and have revised the sentence for clarity. The corrected version now reads: "Object detection is usually inputs the image to be detected into the detection model. Then, it uses rectangular boxes to display the position of interesting objects in the original image and determine their category. Unlike object detection, the MOT algorithm needs not only to output the location of the target of interest, but also to associate the target's identity with the location information output by the detection task and maintain the state of association between the identity information and the bounding box." We have carefully reviewed this manuscript and made revisions to the unclear logical sentences, as well as refined the language expression to ensure that the content is clear, understandable and well-structured. We appreciate your suggestions and believe the revisions have improved the readability of the manuscript. 

Round 2

Reviewer 1 Report

Comments and Suggestions for Authors

The authors have revised their manuscript regarding all the comments made in the review, except the one about including a separate conclusion section after the discussion section. So I urge them to amend the paper in this sense. 

Author Response

Comments 1: The authors have revised their manuscript regarding all the comments made in the review, except the one about including a separate conclusion section after the discussion section. So I urge them to amend the paper in this sense.

Response 1:  Thank you for pointing this out. We agree with this comment. Therefore, we have added a separate conclusion section after the discussion part. This section summarizes the entire content and points out the directions for future research. All the revised contents have been marked in red in the revised version.

Reviewer 2 Report

Comments and Suggestions for Authors

The authors addressed all my suggestions.

Author Response

Thank you very much for your valuable feedback and for taking the time to review our manuscript. We are grateful that you found our revisions satisfactory and appreciate your constructive suggestions, which have significantly improved the quality of the paper.

Reviewer 4 Report

Comments and Suggestions for Authors

Page 3, str. 112 "1. We introduce GC attention to enhance the detection ability of the network." GC ? - This term must be explained before its first use, not after, as on str. 197 on page 5  just now - "GC" is the abbreviation of "Global Context Block".

Author Response

Comments 1: Page 3, str. 112 "1. We introduce GC attention to enhance the detection ability of the network." GC ? - This term must be explained before its first use, not after, as on str. 197 on page 5  just now - "GC" is the abbreviation of "Global Context Block".

Response 1:  Thank you for pointing this out. We agree with this comment. Therefore, we have revised the text on page 3 (line 112) to explicitly define "GC" as "Global Context" upon its first appearance. All the revised contents have been marked in red in the revised version.